# Repeated Oral Administration of a KDEL-Tagged Recombinant Cholera Toxin B Subunit Effectively Mitigates DSS Colitis despite a Robust Immunogenic Response

**DOI:** 10.3390/toxins11120678

**Published:** 2019-11-20

**Authors:** Joshua M. Royal, Micaela A. Reeves, Nobuyuki Matoba

**Affiliations:** 1James Graham Brown Cancer Center, Center for Predictive Medicine, University of Louisville School of Medicine, Louisville, KY 40202, USA; joshua.royal@louisville.edu; 2Department of Pharmacology and Toxicology, University of Louisville School of Medicine, Louisville, KY 40202, USA; micaela.reeves@louisville.edu

**Keywords:** cholera toxin B subunit, DSS colitis, IBD, immunogenicity

## Abstract

Cholera toxin B subunit (CTB), a non-toxic homopentameric component of *Vibrio cholerae* holotoxin, is an oral cholera vaccine antigen that induces an anti-toxin antibody response. Recently, we demonstrated that a recombinant CTB variant with a Lys-Asp-Glu-Leu (KDEL) endoplasmic reticulum retention motif (CTB-KDEL) exhibits colon mucosal healing effects that have therapeutic implications for inflammatory bowel disease (IBD). Herein, we investigated the feasibility of CTB-KDEL for the treatment of chronic colitis. We found that weekly oral administration of CTB-KDEL, dosed before or after the onset of chronic colitis, induced by repeated dextran sodium sulfate (DSS) exposure, could significantly reduce disease activity index scores, intestinal permeability, inflammation, and histological signs of chronicity. To address the consequences of immunogenicity, mice (C57BL/6 or C3H/HeJ strains) were pre-exposed to CTB-KDEL then subjected to DSS colitis and CTB-KDEL treatment. While the pre-dosing of CTB-KDEL elicited high-titer anti-drug antibodies (ADAs) of the immunoglobin A (IgA) isotype in the intestine of C57BL/6 mice, the therapeutic effects of CTB-KDEL were similar to those observed in C3H/HeJ mice, which showed minimal ADAs under the same experimental conditions. Thus, the immunogenicity of CTB-KDEL does not seem to impede the protein’s mucosal healing efficacy. These results support the development of CTB-KDEL for IBD therapy.

## 1. Introduction

The principal virulence factor of *Vibrio cholerae* is cholera toxin (CT), which is comprised of a toxic A subunit and a non-toxic homopentameric B subunit (CTB). Recently, we have shown that a recombinant variant of CTB containing a KDEL endoplasmic reticulum (ER) retention motif (CTB-KDEL) has the ability to induce mucosal healing, facilitate colon epithelial wound repair, and enhance recovery from an acute dextran sulfate sodium (DSS) colitis model in mice [1,2]. These effects were attributed to the addition of the C-terminal ER retention motif, KDEL, to CTB. The KDEL sequence enabled CTB-KDEL to bind to the KDEL receptor and localize within the ER in the Caco2 human colon epithelial cell line. Upon ER localization, CTB-KDEL induced an unfolded protein response (UPR) and subsequent TGFβ signaling. In particular, the inositol-requiring enzyme 1 (IRE1)/X-box binding protein 1 (XBP1) arm of the UPR was indispensable for wound healing activity [2]. These findings were corroborated in primary mouse colon epithelial cells, where CTB-KDEL induced an UPR, while CTB and the non-GM1 binding mutant G33D-CTB-KDEL failed to exhibit such an effect. Moreover, in an ex vivo experiment using colectomy tissue from inflammatory bowel disease (IBD) patients, CTB-KDEL, but not CTB, induced an UPR, upregulated wound healing pathways, and maintained viable crypts [2]. These findings provide implications for the potential use of CTB-KDEL in the treatment of IBD.

It is estimated that 1.3% of US adults (3.1 million) suffer from IBD, which consists of two main classes, ulcerative colitis (UC) and Crohn’s disease (CD). IBD is characterized by chronic periods of remission and relapse [3]. UC mainly affects the mucosa in the rectum and colon, whereas the inflammation in CD affects all layers of the bowel wall including the muscularis and serosa. Moreover, unlike UC, inflammation in CD can occur at any part of the gastrointestinal (GI) tract [4]. Multiple studies using immunosuppressive agents have led to a general consensus that healing of the mucosal layer (i.e., mucosal healing) is the most important treatment goal of IBD [5,6,7]. In particular, the most critical component of the mucosal healing in UC, which does not usually involve transmural inflammation, may be epithelial healing/restitution [5], an effect that could be attained by oral administration of CTB-KDEL. Currently, treatment strategies for UC aim to blunt the inflammatory response and establish remission by employing anti-inflammatory and immunomodulatory agents such as 5-aminosalicylic acid, steroids, and thiopurines. However, 20–40% of patients lose response or are nonresponders to these agents, which will lead to the use of biologics (i.e., anti-TNF agents) that may cause more severe adverse reactions, or surgical resection of the colon [8]. Given that nearly half of patients fail to achieve mucosal healing with available medications [9], we hypothesize that CTB-KDEL may provide a novel therapeutic option to achieve the major goal in UC treatment.

Although oral administration of CTB-KDEL has shown great promise in an acute DSS colitis mouse model, it is yet to be determined whether the treatment can reverse chronic colitis after it has been established. Acute DSS-induced colitis typically only incorporates features of innate immunity and acute epithelial injury [10]. In contrast, the chronic and progressive model of colitis induced by repeated exposure to DSS represents a complex interplay between innate immune responses and compromised epithelial barrier function that drives a chronic sustained inflammatory response [10,11,12].

Another potential issue for the therapeutic use of CTB-KDEL is that the parent molecule, CTB, is a strong mucosal immunogen [13,14,15]. In fact, CTB is used as a component of the World Health Organization (WHO) pre-qualified oral cholera vaccine Dukoral^®^ to stimulate the production of both secretory immunoglobulin A (S-IgA), produced locally in the intestines, and serum immunoglobulin G (IgG) against CT [16,17]. CTB’s ability to induce a potent S-IgA and IgG response upon mucosal administration is attributed to the various cell types within the GI tract that present GM1 ganglioside on their cell surface, such as macrophages, dendritic cells (DCs), B cells, T cells, neurons, and epithelial cells [18,19,20,21]. Previously we have determined that CTB-KDEL’s immunogenicity is equivalent to that of CTB in mice [22]. From a drug development standpoint, the induction of anti-drug antibodies (ADAs) is generally regarded as a risk because they could provoke clinical complications that arise from altered pharmacokinetics, pharmacodynamics, and/or immunotoxicity [23,24]. However, it is yet to be determined whether immunogenicity is a relevant concern for orally administered biologics, as there are very few such agents that have undergone clinical investigation on this matter.

Here, using the repeated DSS exposure mouse colitis model, we investigated whether CTB-KDEL could alleviate chronic colitis and whether the therapeutic effects of CTB-KDEL are altered by the protein’s immunogenicity.

## 2. Results

### 2.1. CTB-KDEL’s Therapeutic Effects in a Mouse Colitis Model Exposed to Repeated DSS Cycles

To characterize the therapeutic effect of CTB-KDEL under chronic colitis conditions, we performed a chronic DSS colitis experiment in mice in which the animals were exposed to three cycles of seven-day DSS exposure followed by 14 days of normal drinking water over 63 days. In this chronic DSS-induced colitis model, mice develop chronic inflammation in the colon after the first cycle of DSS exposure [25]. Thus, mice were treated orally with 3 µg of CTB-KDEL or vehicle control dosed once a week starting at Day 7 (QW^d7^; before chronic colitis had developed), or dosed every three days or QW at Day 35 (Q3D^d35^ and QW^d35^, respectively) (Figure 1). As shown in Figure 2A, CTB-KDEL QW^d7^ administration significantly decreased disease activity index (DAI) scores immediately following the first dose and maintained significantly lower DAI values than the vehicle control-dosed group throughout the study despite exposure to two additional DSS cycles. Mice that were treated QW^d35^ or Q3D^d35^ (dosing started after chronic colitis was established) also showed immediate reduction in DAI scores following CTB-KDEL administration. On Day 63 mice were administered fluorescein isothiocyanate (FITC)-dextran by gavage 4 h prior to euthanasia to test the intestinal permeability of each group. Chronic DSS-induced colitis is known to induce barrier dysfunction followed by increased intestinal permeability that is correlated with worsening of the disease [11]. Compared to mice treated with vehicle control (PBS), those that received QW^d7^ and Q3D^d35^ CTB-KDEL treatment showed significantly lower plasma FITC-dextran levels, indicating that CTB-KDEL treatment improved intestinal barrier function (Figure 2B). QW^d35^ treatment also appeared to have improved the barrier function; however, the decrease in FITC-dextran plasma levels was not statistically significant. Likewise, with QW^d7^ and Q3D^d35^, but not with QW^d35^ dosing, CTB-KDEL significantly prevented colon shrinkage induced by DSS insult (Figure 2C).

To dissect the therapeutic effects of CTB-KDEL, we measured pro-inflammatory cytokine levels in the distal colon tissue lysates at the time of euthanasia (Day 63). Consistent with other reports of chronic colitis [26,27], two key markers of chronic colitis IL-1β and TNFα were significantly elevated in PBS-treated mice when compared to healthy animals (Figure 2D). In contrast, none of the CTB-KDEL treatment groups demonstrated an elevation of the pro-inflammatory cytokines tested; IL-1β, TNFα, GM-CSF, and IFNγ, suggesting an overall suppression of inflammation by CTB-KDEL treatment. In particular, QW^d7^ treatment showed significantly lower GM-CSF and IL-1β levels compared to mice treated with PBS + DSS, while QW^d35^ mice had significantly reduced GM-CSF and IFNγ cytokine levels compared to mice treated with PBS + DSS. Although Q3D^d35^ dosing did seem to reduce pro-inflammatory cytokine levels, these reductions were not statistically significant.

To further corroborate the above results, we performed a histopathological examination and evaluated the presence of active chronic colitis markers, particularly crypt structural alterations and expansion of the lamina propria, in hematoxylin and eosin (H&E)-stained distal colon tissues. By these metrics, CTB-KDEL-treated mice showed increased recovery from chronic DSS-induced colitis compared to vehicle control. QW^d7^ and Q3D^d35^ dosing demonstrated similar but noteworthy recovery, exhibiting fewer instances of crypt branching, crypt distortion, and expansion of the lamina propria than QW^d35^ dosing (Figure 3A). It was indeterminable whether QW^d7^ or Q3D^d35^ dosing had the greatest impact on recovery from chronic colitis. QW^d35^ dosing, however, still showed signs of recovery compared to vehicle control, as the latter displayed the most severe expansion of the lamina propria and crypt alterations (Figure 3A). These histopathological findings were quantified by a crypt score rating of the H&E-stained distal colon tissues. As shown in Figure 3B, all CTB-KDEL-treated groups had a significantly reduced crypt score, which was characterized by a reduction in the shortening or loss of basal crypts and epithelial surface retention in all CTB-KDEL-treated mice, in contrast to PBS treatment (Figure 3B).

### 2.2. The Impact of CTB-KDEL’s Immunogenicity on Its Therapeutic Effect in Acute DSS Colitis Models

Previously, we have shown that oral administration of 3, 10, or 30 µg CTB-KDEL twice at a two-week interval induced anti-toxin serum IgG and fecal IgA antibodies with high titers [22]. Notwithstanding, the results in the chronic DSS colitis study shown in Figure 2 and Figure 3 demonstrate that repeated dosing with 3 µg of CTB-KDEL (eight doses total) did not lose efficacy over the course of the study, raising an interesting question about the consequences of an immune response to the protein. Thus, to address this immunogenicity question, specifically, the induction of ADAs, we used a mouse model of acute DSS colitis in which mice were orally administered with 30 µg of CTB-KDEL twice prior to 3% DSS exposure to induce an immunogenic response to the protein (the aforementioned vaccine dosing regimen [22]). At the end of DSS exposure, mice were orally administered 3 µg of CTB-KDEL; this single therapeutic dose has proven to be efficacious in C57BLl/6J mice [2,22]. The dosing regimens and groups are depicted in Figure 4.

As shown in Figure 5A, after DSS exposure was halted all three CTB-KDEL dosing regimens showed a significantly more rapid recovery from DSS-induced weight loss when compared to PBS-treated mice. Vaccinated + therapeutic + DSS mice showed a significant improvement one day after therapeutic dosing (Day 8), while therapeutic + DSS and vaccinated + DSS mice showed a significant improvement in weight recovery starting at Day 10 and continued throughout the study. These results were confirmed by colon length and DAI scores that were examined upon euthanasia at Day 14. The colon lengths of both groups therapeutically treated with CTB-KDEL on Day 7 were significantly longer than those of the PBS + DSS group (Figure 5B). Additionally, all CTB-KDEL-treated mice showed similar DAI scores, which were significantly lower than those of PBS + DSS mice (Figure 5C). To determine the presence of ADAs in the GI tract of these mice, we performed an ELISA detecting CTB-binding IgAs in feces collected before (Day 7) and after (Day 14) the therapeutic dose of CTB-KDEL was administered. Both vaccinated + DSS and vaccinated + therapeutic + DSS mice had similar and high levels of anti-CTB-KDEL IgA antibodies in the intestine on both days, indicating that ADAs were present when CTB-KDEL was administered for the treatment of colitis (Figure 5D).

To further reveal the impacts of ADAs, we performed the same acute DSS colitis experiment in C3H/HeJ mice which, unlike the C57BL/6 strain, are poor responders to CTB immunization due to the lack of an MHC haplotype reactive to CTB epitopes [28,29]. The experiment was performed exactly as described above (Figure 4) and the same dosing groups were implemented. Similar to the results obtained in C57BL/6 mice, all three CTB-KDEL dosing regimens proved to be efficacious against DSS colitis in contrast to PBS-treated mice (Figure 6). A significant prevention of body weight loss was noted in vaccinated + DSS as early as Day 6, while vaccinated + therapeutic + DSS and therapeutic + DSS mice demonstrated a significant recovery from body weight loss starting on Day 8 (Figure 6A). Furthermore, all CTB-KDEL-treated mice showed significantly reduced colon shrinkage and lower DAI scores compared to PBS + DSS mice and were comparable between all CTB-KDEL treatment groups (Figure 6B,C). Unlike C57BL/6 mice, however, CH3/HeJ mice had little to no anti-CTB IgA response to the vaccination with CTB-KDEL (Figure 6D). Thus, CTB-KDEL’s mucosal therapeutic effects in this model are unlikely to be dependent on IgA induction in the intestine.

## 3. Discussion

We have previously demonstrated CTB-KDEL’s mucosal healing effects in an acute DSS colitis model in C57BL/6 mice and in colonic explants isolated from IBD patients, as well as the prevention of tumor development in an azoxymethane (AOM)/DSS mouse model of colitis-associated cancer [1,2]. However, its development as a biotherapeutic will need to address unique issues associated with its use. Here we aimed to uncover two key questions surrounding CTB-KDEL’s ability to treat chronic symptoms of UC. Specifically, we addressed whether CTB-KDEL could resolve colitis after the formation of chronic inflammatory conditions in a more clinically relevant model based on repeated DSS exposure. Additionally, we determined the impact of immunogenicity on CTB-KDEL’s therapeutic efficacy, as the protein is a potent inducer of mucosal and systemic antibodies [22]. 

Acute DSS-induced colitis typically only incorporates features of innate immunity and acute epithelial injury [30]. In contrast, the chronic and progressive model of DSS colitis presented in this manuscript represents a complex interplay between innate immune responses and compromised epithelial barrier function that drives a chronic sustained inflammatory response [10]. The inflammatory response is driven mainly through the paracellular route created by DSS exposure that is exacerbated by proinflammatory cytokines, such as TNFα and INFγ [31,32]. After repeated DSS cycles, this proinflammatory response leads to a positive feedback loop of increased intestinal permeability, initially driven by DSS, and increased production of inflammatory cytokines in response to bacteria that cross the intestinal barrier and inflammatory cytokines that also cause direct damage to the intestinal epithelium [30,33,34]. Apoptosis of intestinal epithelial cells and subsequent damage to intestinal mucosa magnifies an increase in intestinal permeability, further adding to the positive feedback loop between increased intestinal permeability, enhanced production of inflammatory cytokines, and damaged intestinal epithelium. For these reasons, we initiated oral CTB-KDEL treatment at two different timepoints during the nine-week chronic DSS colitis experiment. One group received the first CTB-KDEL dose at the end of the first DSS cycle (i.e., Day 7) when the DSS--induced injury was in the acute phase, whereas two other groups initiated CTB-KDEL treatment after the second cycle of DSS (i.e., Day 35) when more severe colitis had started to develop. The chronic inflammation and histopathogical findings seen in the vehicle control group are similar to the features of those seen in human UC [10,35], demonstrating the relevance of this model for the purpose of the present study. The data revealed that QW^d7^ and Q3D^d35^ dosing of 3 µg CTB-KDEL were more efficacious than QW^d35^ dosing (Figure 2). Although least effective among the three CTB-KDEL treatment groups, the QW^d35^ dosing group still showed efficacy, as evident from significantly low DAI values compared to the untreated group throughout the last DSS exposure cycle. Meanwhile, dosing every two weeks was not as effective as QW dosing regardless of the timing of initial treatment (data not shown). The therapeutic efficacy of CTB-KDEL against chronic colitis was corroborated by histological evidence for significant remission and healing in mice treated with the protein. In addition, it was also noted that there was minimal neutrophil infiltration in tissues of CTB-KDEL-treated mice in contrast to vehicle control (Figure 3). These results add to the clinical relevance of the present study because histologic remission and healing is an area of increased research focus and holds the promise of being an important marker of treatment efficacy in UC [36,37]. Of note, all CTB-KDEL-dosed groups showed reduction in intestinal permeability (serum FITC-dextran levels), downregulation of pro-inflammatory cytokine levels and decreased histological damage in the colon, all three of the main hallmarks of IBD [26,30,38].

The significance of the above results is twofold. First, the results indicate that the animals continued to respond to the medication over time. Conversely, CTB-KDEL did not lose its efficacy with repeated dosing (eight doses total for QW^d7^, four doses for QW^d35^, and nine doses for Q3D^d35^). Second, as the repeated DSS model usually causes chronic inflammation in the colon after the first cycle of DSS exposure [25], the positive results in the delayed dosing groups indicate that CTB-KDEL is effective against chronic colitis. Collectively, these data provide a basis for both QW and Q3D^d35^ dosing to be used in the treatment of chronic colitis; QW dosing may be sufficient as a maintenance dose to prevent relapse of disease, while Q3D dosing may be better suited for treatment of active disease.

A key pharmacological and toxicological question for biotherapeutic development is immunogenicity; specifically, ADA induction. A major disadvantage of therapeutic proteins is that almost all induce an antibody response [39]. ADAs constitute a theoretical risk because they may lead to a loss of efficacy, altered pharmacokinetics, and potentially cause immunotoxicity [23,24,40,41]. As CTB-KDEL is a derivative of the cholera vaccine antigen CTB, it is of critical importance to assess the consequences of an immune response to CTB-KDEL for its development as a drug for chronic diseases like UC, which would likely need repeated and long-term medication. Since nonclinical models of immunogenicity are not always predictive of immune responses in humans [41], we used two strains of mice with different H-2 haplotypes to help understand the dynamics of CTB-KDEL’s immunogenicity on its therapeutic effects in colitis. To that extent, the study using C57BL/6 mice (*H-2^b^* haplotype) represented a scenario of high ADA induction, while C3H/HeJ mice (*H-2^k^* haplotype) provided an alternative scenario of minimal ADA induction. In the experiment using C57BL/6 mice, pre-dosing of the animals twice with CTB-KDEL (i.e., “vaccinated”) indeed induced significant levels of anti-CTB-KDEL IgA antibodies in the intestine before DSS exposure and throughout the rest of the study (Figure 5D). Although we did not directly analyze the capacity of the ADAs to neutralize CTB-KDEL’s effect, the therapeutic efficacy observed in vaccinated + therapeutic + DSS group was not inferior to that of therapeutic + DSS group, suggesting that intestinal ADAs arising from repeated CTB-KDEL administrations have limited impacts, if any, on the protein’s mucosal healing efficacy. However, interpretation of the results from this experiment is complicated by the fact that the CTB-KDEL vaccination also showed marked prophylactic effects against DSS colitis. Our preliminary study shows that CTB-KDEL is only detectable in the colon up to 48 h after a single oral administration in CTB-KDEL-naïve mice (data not shown). Thus, we initially speculated that the prophylactic effect of CTB-KDEL seen in C57BL/6 mice could be due to the ADAs extending the bioavailability of the protein in the colon. Alternatively, the induction of ADAs itself might have played a beneficial role in mitigating inflammation because IgA is known to help maintain mucosal homeostasis and mediate anti-inflammatory functions [42,43,44,45,46]. In fact, this was previously demonstrated in an experimental model of asthma where CTB alleviated allergic inflammation via the induction of a secretory IgA response [47]. However, both of the above hypotheses are rejected by the results observed in the C3H/HeJ mouse experiment (Figure 6), which showed a similar prophylactic effect of CTB-KDEL in spite of few ADAs induced due to the lack of an MHC II haplotype capable of recognizing CTB peptides in the mouse strain [29].

At present, it remains unknown how predosing of CTB-KDEL showed a profound prophylactic effect against DSS acute colitis. While our current premise is that epithelial restitution is the major contributing function of CTB-KDEL’s therapeutic effect in colitis [2], we cannot discount the potential role of the immune system. It has been demonstrated previously that CTB can directly modulate immune cells through binding to their cell-surface GM1 ganglioside. For example, Rouquette-Jazdanian et al. showed that the binding of CTB to GM1 ganglioside prevented the activation and proliferation CD4+ T cells [48]. Coccia et al. determined that CTB inhibited mucosal Th1 cell signaling and Th1 cytokine production of colonic lamina propria mononuclear cells, which resulted in the mitigation of 2,4,6-trinitrobenzene sulfonic acid-induced mouse model of Crohn’s disease [49]. Furthermore, D’Ambrosio et al. demonstrated CTB promotes Treg cells by preventing monocyte-derived DC maturation [50]. It remains to be determined whether CTB-KDEL exhibits such direct immunomodulatory effects in immune cells as those demonstrated for CTB. Future studies will need to address the impact of CTB-KDEL on the gut immune system to uncover its potential immunomodulatory function in the context of colitis. Nevertherless, the present study demonstrated that CTB-KDEL is efficacious against DSS colitis regardless of ADA levels in the gut. Combined with the significant efficacy observed with repeated CTB-KDEL administration in the chronic DSS colitis model (Figure 2 and Figure 3), our data strongly suggest that the consequences of CTB-KDEL’s immunogenicity on its ability to treat UC may be negligible.

## 4. Conclusions

In conclusion, the present study provides evidence for CTB-KDEL’s ability to treat chronic colitis. Additionally, we argue that immunogenicity is not a major risk for the mucosal healing activity of orally administered CTB-KDEL. Further research is needed to determine why ADAs did not neutralize or prevent CTB-KDEL’s activity and to determine whether this phenomenon is unique to CTB-KDEL or applies to other GI-targeted biologics. Nevertheless, collectively these results support the development of CTB-KDEL for UC therapy.

## 5. Materials and Methods

### 5.1. Animals

Eight-week-old C57BL/6J, female mice were obtained from Jackson Laboratories (Bar Harbor, ME, USA). The University of Louisville’s Institutional Animal Care and Use Committee approved all animal studies conducted in this manuscript. (IACUC 18245 approved on 25 April 2018)).

### 5.2. CTB-KDEL

CTB-KDEL was produced in *Nicotiana benthamiana* and purified to >95% homogeneity with an endotoxin level of 0.05–0.1 endotoxin units per mg, as described previously [22,51] but with modification in the final chromatography step to selectively purify the proteins with fully intact C-terminus (Morris et al. manuscript in preparation). Purity and pentamer formation were assessed by SDS-PAGE under denaturing and non-denaturing conditions, whereas the molecular weight of CTB-KDEL was verified by mass spectrometry, as described before [22].

### 5.3. Chronic DSS Colitis Model

Groups of 10 female eight-week-old mice, randomly assigned, were used. Mice were exposed to 2% DSS (MW 36,000–50,000; MP Biomedicals, Solon, OH, USA) for seven days followed by 14 days of normal drinking water. The DSS exposure and recovery cycle was repeated three times and mice were sacrificed following the third cycle on Day 63. Disease activity index (DAI) scores, consisting of body weight loss, fecal consistency, and occult blood tests, were recorded daily and performed as previously described [11].

### 5.4. Intestinal Permeability Analysis

Intestinal permeability was determined by implementing a modified method using fluorescein isothiocyanate-dextran (FITC-dextran) (FD4, MS 3000–5000; Sigma Aldrich, Milwaukee, WI, USA) [52]. On Day 62, mice were fasted overnight but allowed free access to water. The next morning (Day 63), mice were administered with FITC-dextran diluted in PBS, 100 mg/mL, (0.6 mg/g body weight) by oral gavage. After 4 h, whole blood was collected, and serum was isolated. Serum was diluted with equal volume of PBS and tested in duplicate for fluorescence measurement by a microplate reader (BIotek Synergy H1, Winooski, VT, USA). The concentration of FITC-dextran was determined using a standard curve generated by dilutions of FITC-dextran in PBS (0, 125, 250, 500, 1000, 2000, 4000, 6000, 8000 ng/mL).

### 5.5. Protein Isolation and Quantification

Distal colon tissue isolated at euthanasia was snap frozen in liquid nitrogen and pulverized with a Bessman Tissue Pulverizer and placed in T-PER (Thermo Scientific, Rockfordm, IL, USA) with a protease inhibitor cocktail (Sigma-Aldrich, St. Louis, MO, USA). Total protein was isolated by gravity centrifugation, 10,000× *g* for 5 min, of tissue fragments followed by collection of the buffer containing isolated protein, and storage at −80 °C until analysis. Protein concentrations were determined using a bicinchoninic acid (BCA) protein assay (Sigma-Aldrich, Milwaukee, WI, USA) according to the manufacturer’s protocol and normalized for all samples prior to loading on a Mouse Cytokine/Chemokine Magnetic Bead Panel (EMD Millipore, Billerica, MA, USA). The panel was analyzed with a Milliplex MAP Kit on a MagPix with Luminex xMAP technology (Luminex, Northbrook, IL, USA).

### 5.6. Histology

Colons were removed and washed with PBS. A portion of the distal colon was fixed with 10% paraformaldehyde overnight and stored in 70% ethanol until paraffin embedding, sectioning, and routine hematoxylin and eosin (H&E) staining. Crypt scoring was performed as previously described [1]. Tissue sections from 10 mice were scored in a blinded manner and averaged for each group. Histopathological examination of hematoxylin and eosin (H&E)-stained distal colon tissue sections was performed in a blind manner.

### 5.7. Acute DSS Colitis Models to Study the Impact of CTB-KDEL Immunogenicity

Groups of 10 female C57BL/6 or C3H/HeJ mice, randomly assigned, were used. Animals were gavaged with PBS or 30 μg CTB-KDEL twice at a two-week interval after sodium bicarbonate administration, as described previously [22]. Two weeks after the second dose of CTB-KDEL, DSS exposure was initiated. Body weights were measured at the initiation of DSS exposure as a baseline and every morning thereafter to determine percent change. Animals received 3% DSS (MW 36,000–50,000; MP Biomedicals, Santa Ana, CA, USA) in drinking water for seven days. On the seventh day of DSS exposure, animals gavaged with 100 µL PBS or 3 µg (0.03 mg/mL solution) CTB-KDEL after sodium bicarbonate (200 µL of 30 mg/mL solution) administration, as described previously [1], and allowed to recover for seven days during which the animals received normal drinking water.

### 5.8. Detection of Fecal IgAs

Anti-CTB-specific IgA levels in fecal extracts were determined using a previously described method [22]. Briefly, MaxiSorp ELISA plates (Nalgene Nunc International, Thermo Fisher, Waltham, MA, USA) were coated overnight with 1 µg/mL of CTB-KDEL. After blocking, 100 µL of fecal samples in appropriate dilutions were added to the ELISA plates. The IgAs bound to the plates were detected using goat anti-mouse IgA antibodies conjugated with horseradish peroxidase (HRP; SouthernBiotech, Birmingham, AL, USA) and a TMB Super Sensitive One Component HRP Microwell Substrate (SurModics BioFx). Absorbance was measured at 450 nm with a microplate reader (BIotek Synergy H1, Winooski, VT, USA) after the reaction was stopped. Dilutions of purified mouse IgA standards were used to optimize the ELISA. The titer was defined as the greatest dilution factor of the sample with positive OD450 reading after subtracting an average background value.

### 5.9. Statistics

For all data, outliers were determined by statistical analysis using the Grubb’s test and excluded from further analysis if *p* < 0.05. Graphs were prepared and analyzed using Graphpad Prism version 5.0 (Graphpad Software, La Jolla, CA, USA). To compare two data sets, an unpaired, two-tailed Student’s *t* test was used. To compare three or more data sets, one-way ANOVA with Bonferroni’s multiple comparison post-test or Kruskal–Wallis test with Dunn’s multiple comparison post-test were performed. For body weights and DAI results, a two-way ANOVA with Bonferroni’s multiple comparison post-test was employed.

## Figures and Tables

**Figure 1 toxins-11-00678-f001:**
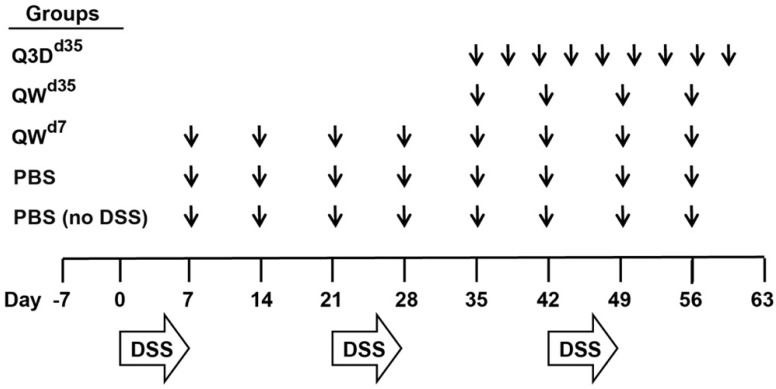
Dextran sodium sulfate (DSS) chronic colitis study design. Mice were exposed to three cycles of DSS that consisted of 2% DSS for seven days followed by 14 days of normal drinking water. Mice were treated orally with 3 µg of cholera toxin B subunit (CTB)-KDEL or vehicle control; PBS, PBS + DSS, and QW^d7^ mice were dosed once a week starting at Day 7, QW^d35^ mice were dosed once a week starting at Day 35, and Q3D^d35^ mice were dosed every three days starting at Day 35. Disease activity index (DAI) scores were measured daily starting at Day 0. Following euthanasia on day 63, colon lengths were measured, serum was collected for fluorescein isothiocyanate (FITC)-dextran analysis, and distal colon tissues were collected for cytokine quantification and histological examination.

**Figure 2 toxins-11-00678-f002:**
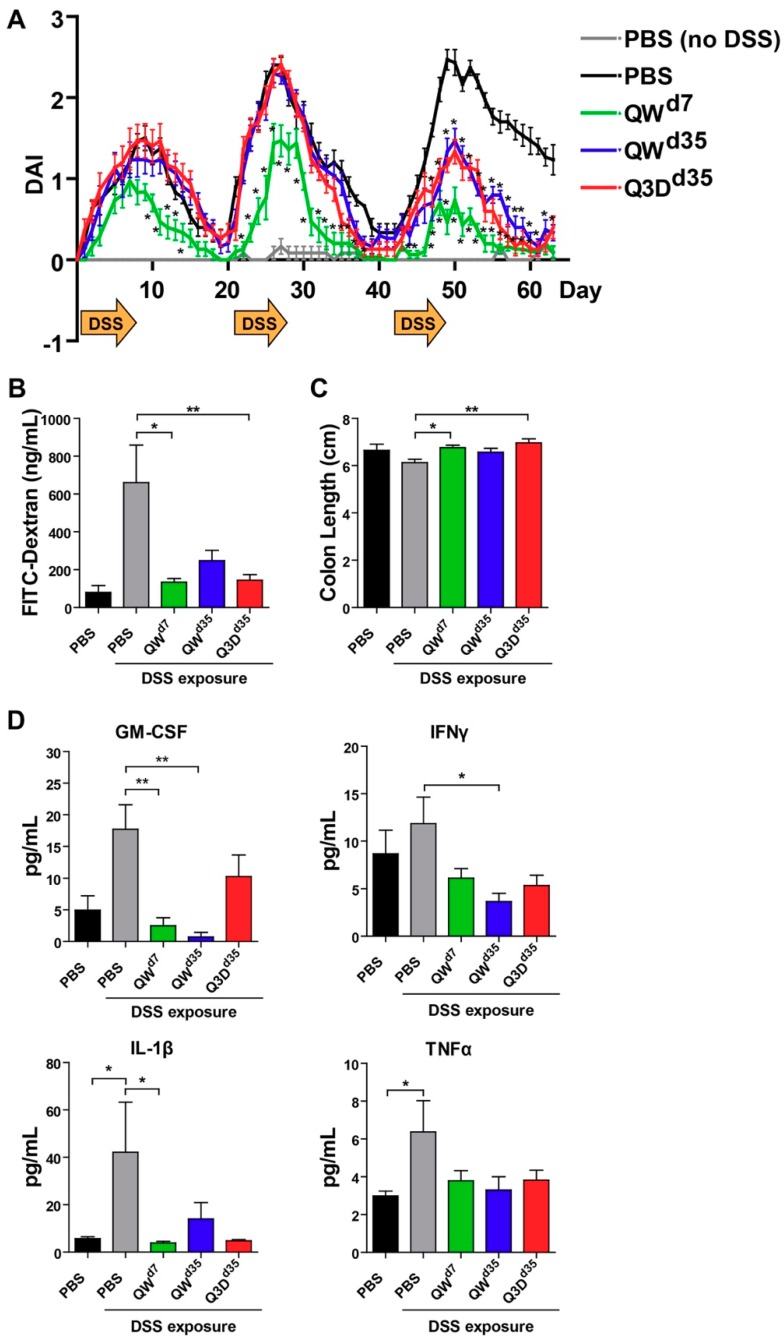
CTB-KDEL mitigates DSS-induced injury and inflammation in a DSS chronic colitis study. C57BL/6 mice (n = 10) were exposed to three cycles of seven-day 2% DSS and 14-day normal drinking water over 63 days and treated with 3 µg CTB-KDEL or PBS dosed orally every week starting at Day 7 (QW^D7^) or starting at Day 35 dosed once a week (QW^D35^) or every three days (Q3D^D35^). (**A**) The time course of disease activity index (DAI) values (mean ± S.E.M.) in each group measured from Days 0–63. * *p* < 0.05; two-way repeated measures ANOVA with Bonferroni’s multiple comparison tests. (**B**) FITC-dextran concentrations found in mice serum 4 h after administration on Day 63. (**C**) Colon length measured after Day 63 sacrifice. (**D**) Proinflammatory cytokines in distal colon tissues were measured by Luminex after Day 63 sacrifice. (**B**–**D**) * *p* < 0.05, ** *p* < 0.01; one-way repeated measures ANOVA with Bonferroni’s multiple comparison tests.

**Figure 3 toxins-11-00678-f003:**
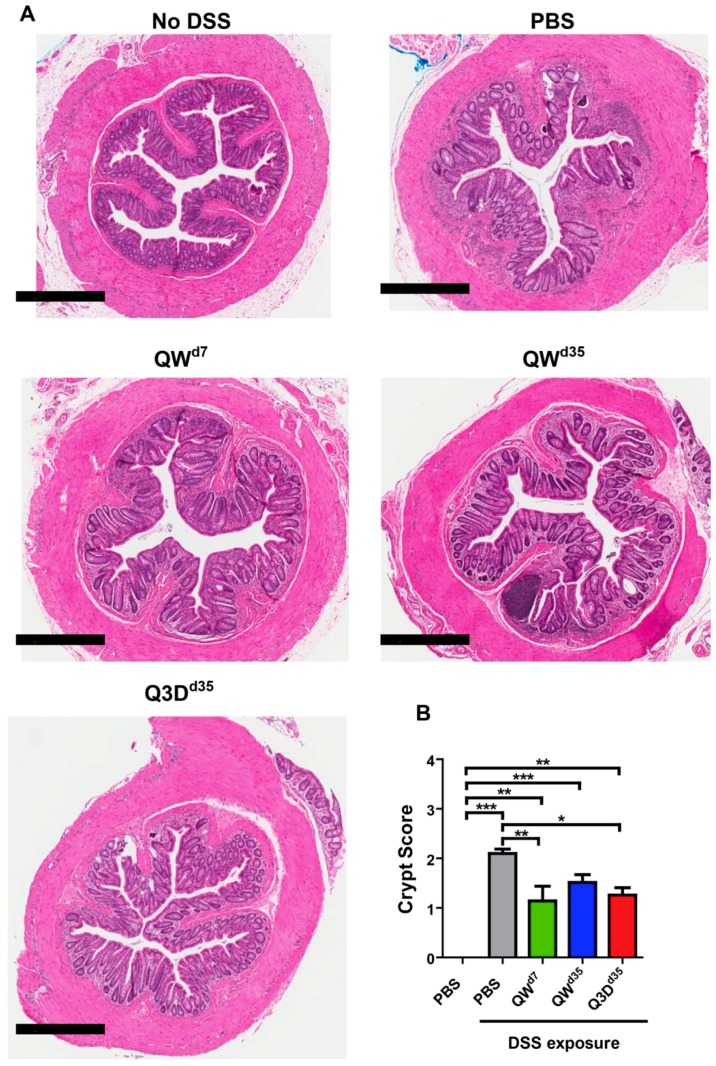
Histological findings from the DSS chronic colitis model. CTB-KDEL treatment protected mice from inflammation and loss of colonic epithelial surface and crypts. (**A**) Representative 4× photomicrographs of hematoxylin and eosin (H&E)-stained distal colon tissues from mice treated with 3 µg CTB-KDEL or PBS dosed orally every week starting at Day 7 (QW^D7^) or starting at Day 35 dosed once a week (QW^D35^) or every three days (Q3D^D35^) starting at Day 35. Scale bars = 600 µM. (**B**) Colon crypt scoring from paraffin-embedded tissue sections were scored after H&E staining. Scoring was based on a 0–4 scale. * *p* < 0.05, ** *p* < 0.01, *** *p* < 0.001; one-way repeated measures ANOVA with Bonferroni’s multiple comparison tests.

**Figure 4 toxins-11-00678-f004:**
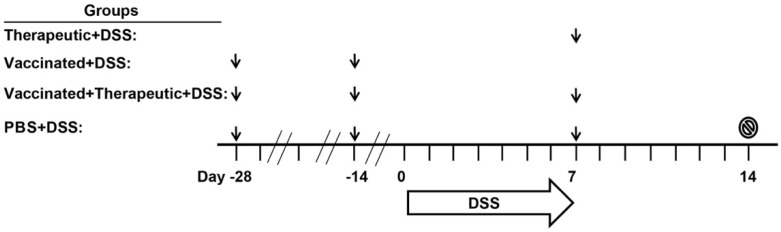
Immunogenicity and acute DSS colitis study design. Four groups of mice were studied; PBS (healthy vehicle control), PBS + DSS (vehicle control in diseased mice), vaccinated + DSS, therapeutic + DSS, and vaccinated + therapeutic+DSS. All mice except for Therapeutic + DSS mice were vaccinated with 30 µg CTB-KDEL or PBS 28 and 14 days (Days −28 and −14) prior to DSS administration. On Day 0, mice were given 3% DSS in drinking water for seven consecutive days. On the seventh day of DSS exposure, DSS was removed from the drinking water and then PBS, PBS + DSS, therapeutic + DSS, and vaccinated + therapeutic + DSS mice were administered with 3 µg CTB-KDEL or a vehicle control (PBS) in a volume of 100 µL after sodium bicarbonate (200 µL of 30 mg/mL solution) administration. Seven days later, on Day 14, mice were euthanized (stop sign) and colon lengths and a disease activity index scores were measured. Body weights were measured daily and feces were collected on Days 7 and 14 from each mouse. Black arrows represent the timings of oral administration of CTB-KDEL or PBS.

**Figure 5 toxins-11-00678-f005:**
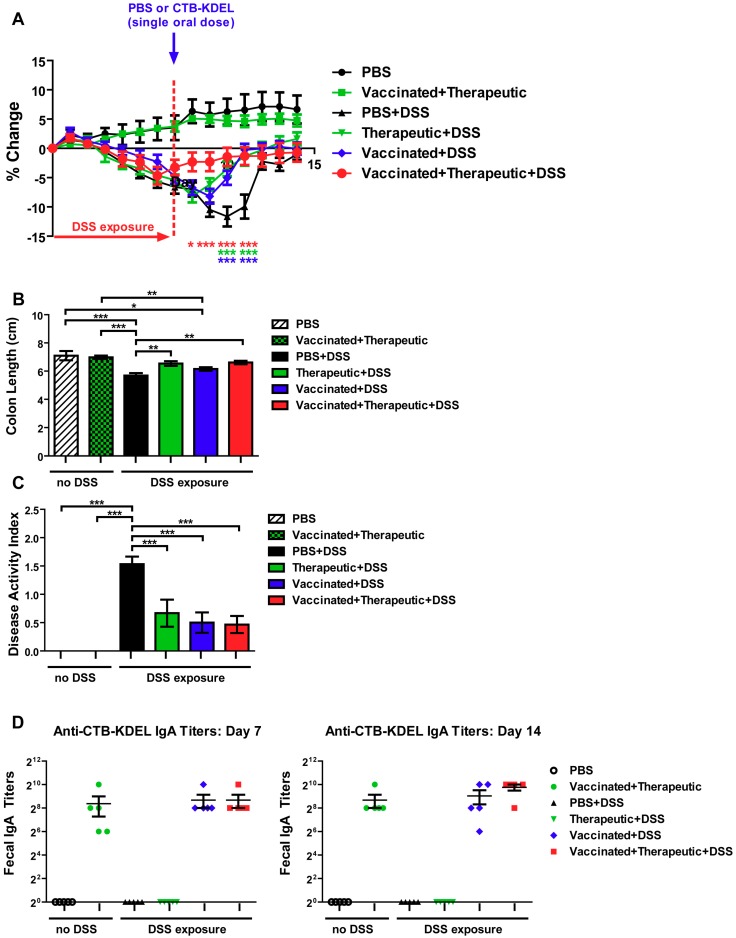
The impact of a pre-existing immune response to CTB-KDEL on its therapeutic efficacy against DSS acute colitis in C57BL/6 mice. C57BL/6 mice were pre-dosed twice orally with 30 µg CTB-KDEL or PBS on Day −28 and Day −14 and then exposed to DSS on Days 0–7. A therapeutic dose of CTB-KDEL (3 µg) was dosed orally on Day 7. (**A**) Percent body weight change was monitored daily. Two-way ANOVA with Bonferroni’s multiple comparison tests were used to compare groups. * *p* < 0.05, ** *p* < 0.01, *** *p* < 0.001, compared to PBS + DSS group. (**B**) Colon length measured at sacrifice on Day 14. (**C**) Disease activity index (DAI) at sacrifice on Day 14. * *p* < 0.05, ** *p* < 0.01, *** *p* <0.001. One-way ANOVA with Bonferroni’s multiple comparison tests. (**D**) Fecal anti-CTB-KDEL IgA titers were determined by ELISA on Days 7 (**left**) and 14 (**right**).

**Figure 6 toxins-11-00678-f006:**
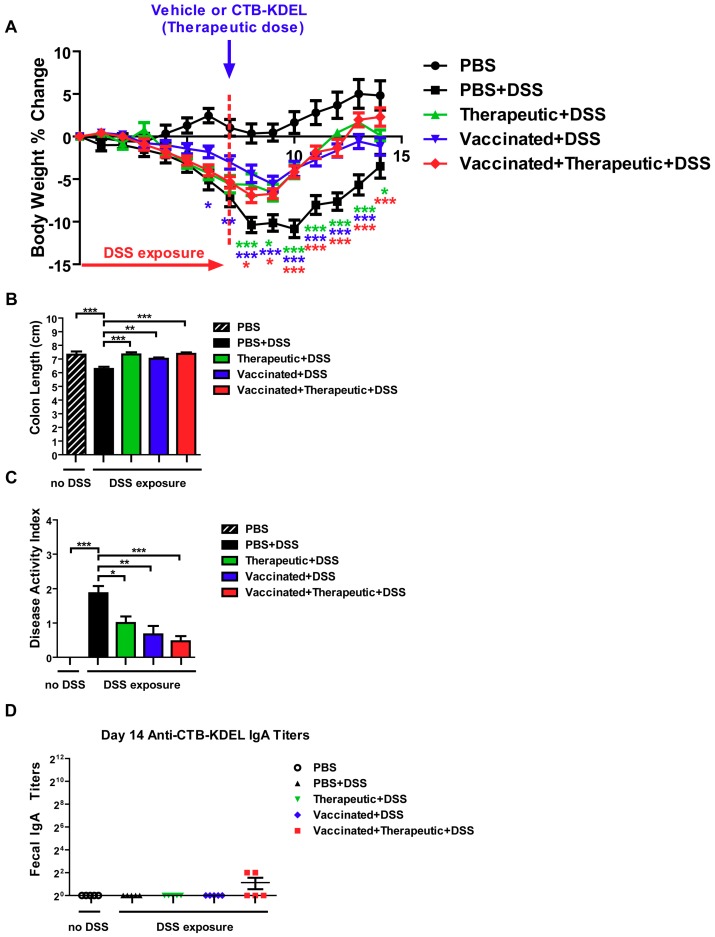
The impact of CTB-KDEL predosing on its therapeutic efficacy against DSS acute colitis in C3H/HeJ mice. C3H/HeJ mice were pre-dosed twice orally with 30 µg CTB-KDEL or PBS on Day −28 and Day −14 and then exposed to DSS on Days 0–7. A therapeutic dose of CTB-KDEL (3 µg) was dosed orally on Day 7. (**A**) Percent body weight change was monitored daily. Two-way ANOVA with Bonferroni’s multiple comparison tests were used to compare groups. * *p* < 0.05, ** *p* < 0.01, *** *p* < 0.001, compared to PBS + DSS group. (**B**) Disease activity index (DAI) at sacrifice on Day 14. (**C**) Colon length measured at sacrifice on Day 14. * *p* < 0.05, ** *p* < 0.01, *** *p* < 0.001. One-way ANOVA with Bonferroni’s multiple comparison tests. (**D**) Fecal anti-CTB-KDEL IgA titers were determined by ELISA on Day 14.

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
