# Peer review of "Repeated Oral Administration of a KDEL-Tagged Recombinant Cholera Toxin B Subunit Effectively Mitigates DSS Colitis despite a Robust Immunogenic Response"

_toxins, 2019, doi:10.3390/toxins11120678_

Round 1

Reviewer 1 Report

In this manuscript, the experiments were well-designed and well-controlled. You convincingly show that application of a KDEL-tagged recombinant cholera toxin B subunit effectively improves chronic colitis which is induced by DSS. You  have experience with the mouse model, and both the model and the effect of treatment are extensively presented. The aim of the study obviously was to prove the effect of treatment by this recombinant toxin subunit.  The treatment with a KDEL subunit toxin induced a robust IgA-mediated immune response and significantly reduced intestinal lesions and inflammation.

You emphasize that these effects occur simultaneously with the IgA response. However, you do not mention in the paper the well-documented anti-inflammatory effect of IgA antibodies. You detected at the site of reduced inflammatory bowel lesions highly elevated IgA levels in the colon and a reduction of inflammatory cytokines. The anti-inflammatory effect of IgA antibodies should be discussed ( please see below for further references).

The Profile of Human Milk Metabolome, Cytokines, and Antibodies in Inflammatory Bowel Diseases Versus Healthy Mothers, and Potential Impact on the Newborn.Meng X, Dunsmore G, Koleva P, Elloumi Y, Wu RY, Sutton RT, Ambrosio L, Hotte N, Nguyen V, Madsen KL, Dieleman LA, Chen H, Huang V, Elahi S.J Crohns Colitis. 2019 Mar 30;13(4):431-441. doi: 10.1093/ecco-jcc/jjy186. PMID 30418545 Intestinal IgA as a modulator of the gutOkai S, Usui F, Ohta M, Mori H, Kurokawa K, Matsumoto S, Kato T, Miyauchi E, Ohno H, Shinkura R. Gut Microbes. 2017 Sep 3;8(5):486-492. doi: 10.1080/19490976.2017.1310357. Epub 2017 Apr 6. Review. PMID: 28384049 B Cell-Activating Factor (BAFF)-Targeted B Cell Therapies in Inflammatory Bowel Diseases. Uzzan M, Colombel JF, Cerutti A, Treton X, Mehandru S.Dig Dis Sci. 2016 Dec;61(12):3407-3424. Epub 2016 Sep 21. Review.PMID:2765510 Secretory immunoglobulin A: well beyond immune exclusion at mucosal surfaces. Corthësy B. Immunopharmacol Immunotoxicol. 2009 Jun;31(2):174-9. doi: 10.1080/08923970802438441. Review. PMID: 19514992 IgA, IgA receptors, and their anti-inflammatory Mkaddem SB, Christou I, Rossato E, Berthelot L, Lehuen A, Monteiro RC. Curr Top Microbiol Immunol. 2014;382:221-35. doi: 10.1007/978-3-319-07911-0_10. Review. PMID: 25116102

Author Response

Reviewer general comment: In this manuscript, the experiments were well-designed and well-controlled. You convincingly show that application of a KDEL-tagged recombinant cholera toxin B subunit effectively improves chronic colitis, which is induced by DSS. You have experience with the mouse model, and both the model and the effect of treatment are extensively presented. The aim of the study obviously was to prove the effect of treatment by this recombinant toxin subunit. The treatment with a KDEL subunit toxin induced a robust IgA-mediated immune response and significantly reduced intestinal lesions and inflammation.

Author Response: We appreciate Reviewer’s comprehensive summary and very positive remarks on our paper. Please see our responses to other specific points below. Pages and lines refer to the revised version.

You emphasize that these effects occur simultaneously with the IgA response. However, you do not mention in the paper the well-documented anti-inflammatory effect of IgA antibodies. You detected at the site of reduced inflammatory bowel lesions highly elevated IgA levels in the colon and a reduction of inflammatory cytokines. The anti-inflammatory effect of IgA antibodies should be discussed (please see below for further references).

Author Response: We agree and, in fact, have discussed the anti-inflammatory effect of IgA response in our original manuscript. Please see below, lines 305-309, where we discuss the potential for intestinal IgAs to mediate inflammation in the experimental colitis experiments performed. We have slightly modified the sentence describing Ref 47 to highlight the beneficial effect of an IgA response in the allergic asthma model.

            Discussion, p. 11, ll. 305-309: “Alternatively, the induction of ADAs itself might have played a beneficial role in mitigating inflammation because IgA is known to help maintain mucosal homeostasis and mediate anti-inflammatory functions [42-46]. In fact, this was previously demonstrated in an experimental model of asthma where CTB alleviated allergic inflammation via the induction of a secretory IgA response [47].”

However, our conclusion in the present study is that the anti-inflammatory activity of the IgA response is not primarily responsible for CTB-KDEL’s therapeutic effects, because the protein exhibited similar efficacy in C3H mice in which an IgA response was negligible (ll. 309 – 312).

Reviewer 2 Report

Author performed an interesting study on the  inflammatory bowel disease (IBD) on using the repeated DSS exposure mouse colitis model. They investigated the protective effect of a CTB recombinant tagged protein, CTB-KDEL, on both acute and chronic DSS-induced colitis mouse model, to find out if the treatment can reverse colitis established alterations.

They assessed that CTB-KDEL had therapeutic effects on both forms of IBD, and, moreover, that, also if this recombinant protein could induce antigen anti-drug antibodies (ADA), protein effects were not impaired by the increase of fecal IgA.

In conclusion, they propose CTB-KDEL to treat chronic colitis.

This study is very interesting and well performed.

In the speculation about the possible mechanisms underlying the inflammatory decrease of colitis-induced citokine expression, it would be interesting to evaluate the acetylation level of intestinal tissue DNA. As previously reported, DSS has been shown to increase the histone 4 acetylation on lysine (K8 and K12) in inflamed mucosa as compared with uninflamed mucosa.

-Since CTB-KDEL mitigates DSS induced injury and inflammation, is it involved also in the epigenetic regulation of the IBD by reducing/inhibiting histone deacetylases (HDACs), thus reducing disease severity and the expression of pro-inflammatory cytokines?

Minor suggestions

Methods

Methods do not describe adequately: 1) IgG and IgA determination; 2) Luminex serum determination

Author Response

Reviewer general comment: Author performed an interesting study on the  inflammatory bowel disease (IBD) on using the repeated DSS exposure mouse colitis model. They investigated the protective effect of a CTB recombinant tagged protein, CTB-KDEL, on both acute and chronic DSS-induced colitis mouse model, to find out if the treatment can reverse colitis established alterations. They assessed that CTB-KDEL had therapeutic effects on both forms of IBD, and, moreover, that, also if this recombinant protein could induce antigen anti-drug antibodies (ADA), protein effects were not impaired by the increase of fecal IgA. In conclusion, they propose CTB-KDEL to treat chronic colitis.

This study is very interesting and well performed.

In the speculation about the possible mechanisms underlying the inflammatory decrease of colitis-induced citokine expression, it would be interesting to evaluate the acetylation level of intestinal tissue DNA. As previously reported, DSS has been shown to increase the histone 4 acetylation on lysine (K8 and K12) in inflamed mucosa as compared with uninflamed mucosa.

-Since CTB-KDEL mitigates DSS induced injury and inflammation, is it involved also in the epigenetic regulation of the IBD by reducing/inhibiting histone deacetylases (HDACs), thus reducing disease severity and the expression of pro-inflammatory cytokines?

Author Response: We appreciate Reviewer’s in depth summary and encouraging remarks on our paper. We are delighted by the comment that “This study is very interesting and well performed”. We also find the Reviewer’s insight into the possible mechanism underlying the decrease in inflammation found in CTB-KDEL treated mice to be very thoughtful. We will consider analyzing acetylation levels in inflamed tissues of treated vs non-treated mice in future mouse DSS experiments. Furthermore, we believe that the manuscript has been improved after revision, taking into account the Reviewer’s constructive and valuable comments. Please see our responses to other specific points below. Pages and lines refer to the revised version.

Methods do not describe adequately:

IgG and IgA determination

Author Response: The method used to quantify fecal IgA levels was added to the method section. Please note that IgGs were not quantified in this study because CTB-KDEL is a luminally active protein for which intestinal IgA would be the most relevant ADAs.

Methods, p. 13, ll. 393-403:

5.8. Detection of fecal IgAs

     Anti-CTB specific IgA levels in fecal extracts were determined using a previously described method [22]. Briefly, MaxiSorp ELISA plates (Nalgene Nunc International) were coated overnight with 1 µg/ml of CTB-KDEL. After blocking, 100 µl of fecal samples in appropriate dilutions were added to the ELISA plates. The IgAs bound to the plates were detected using goat anti-mouse IgA antibodies conjugated with horseradish peroxidase (HRP; SouthernBiotech) and a TMB Super Sensitive One Component HRP Microwell Substrate (SurModics BioFx). Absorbance was measured at 450 nm with a microplate reader (BIotek Synergy H1, Winooski, VT) after the reaction was stopped. Dilutions of purified mouse IgA standards were used to optimize the ELISA. The titer was defined as the greatest dilution factor of the sample with positive OD450 reading after subtracting an average background value.

Luminex serum determination

Author Response: We apologize for the wrong figure legend description of the cytokines measured in Figure 2D. As mentioned in the results section (lines 125-128) and the methods section (lines 367-376), the cytokines shown in Figure 2D were analyzed from distal colon tissue lysates. To address this mistake, we have revised the Figure 2 legend as follows:

Figure 2 legend, p. 4, ll. 122-123:Proinflammatory cytokines in distal colon tissues were measured by Luminex after day 63 sacrifice.”